# Experimental Study on the Bending Resistance of Hollow Slab Beams Strengthened with Prestressed Steel Strand Polyurethane Cement Composite

Jin Li [1,*], Yongshu Cui [1], Dalu Xiong [2], Zhongmei Lu [2], Xu Dong [1,*], Hongguang Zhang [2], Fengkun Cui [1] and Tiancheng Zhou [1]

1   School of Transportation Civil Engineering, Shandong Jiaotong University, Jinan 250357, China
2   Jinan Kingyue Highway Engineering Company Limited, Jinan 250220, China
*   Correspondence: sdzblijin@163.com (J.L.); dongxu512@126.com (X.D.)

**Abstract:** In order to explore the toughening performance and failure mechanism of hollow slab beams strengthened with prestressed steel strand polyurethane cement composite, three test beams (L1–L3) were strengthened and one test beam (L0) was used as a comparison. The influence of different tensile stresses of steel strand and fiber additions on the flexural bearing capacity of the hollow slab beams, was studied. The cracking characteristics, load deflection relationship, ductility and strain of each test beam were compared and analyzed. The test results showed that the toughened material was well bonded to the hollow slab beam and the steel strand, which effectively inhibited the development of cracks in the test beams. The flexural bearing capacity of the strengthened test beams was significantly improved. The use of prestressed steel strand polyurethane cement composite material effectively improved the flexural bearing capacity of the test beams, and this reinforcement process can be further extended to engineering applications.

**Keywords:** steel strand polyurethane cement; hollow slab beam; bending test; reinforcement process; material properties

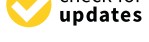



## 1. Introduction

With the rapid growth of traffic volume in China, the vehicle transport load continuously increases. Due to the effects of rain, snow, weathering and other natural conditions, a large number of bridges built at various times in the past have appeared to gradually deteriorate and erode [1–4]. In particular, the prefabricated concrete hollow slab beams put into operation at the end of the 20th century suffer from these conditions [5,6], affecting the normal passage of the road and causing huge economic losses. Although existing reinforcement methods [7–13] could effectively improve the bearing capacity of old bridges, there are still some limitations in controlling the self-weight of the structure, shortening the reinforcement period and simplifying the reinforcement process. In recent years, polyurethane cement composite, a lightweight and high strength corrosion resistant material [14–17], has become a hot research topic in the field of bridge reinforcement.

In recent years, polyurethane cement composite has gradually been applied to practical projects. Due to characteristics such as simplicity, high strength and corrosion resistance, it has become a new type of reinforcement method. Many experts have studied the field of polyurethane reinforcement to a certain extent: Zhou Yonghong [18] proposed a design and construction technology scheme to improve the lateral load distribution and increase the overall stiffness of bridges by using MPC composite reinforcement under the condition of uninterrupted railroad transportation by relying on the load test before reinforcement of the project. They verified the reliability of the reinforcement scheme by establishing a finite element model and performing load tests after reinforcement. Sun Quansheng et al. [19,20] analyzed the reinforcement effects of polyurethane cement wire rope through

tension and compression tests on polyurethane cement and flexural loadbearing damage tests on 3 m ordinary reinforced concrete T-beams with different polyurethane wire rope reinforcement schemes. They verified the practical engineering value of polyurethane wire rope flexural reinforcement, showing that it could effectively improve the flexural loadbearing capacity of the test beams. Wang Jianlin et al. [21], taking the actual project as an example, deduced the theoretical calculation formula of using MPC composite material to reinforce the normal section of a hollow plate girder bridge, and used a super strong, high-toughness MPC composite material to design and construct the hollow plate girder bridge reinforcement. They concluded that their method could effectively improve the ultimate bearing capacity in normal use, and that maintenance and reinforcement could be achieved without interrupting traffic operation.

Based on this, the authors of the present work designed a flexural bearing capacity test for hollow slab beams by using polyurethane cement composite material and steel strand reinforcement. This test was used to explore improvements of the toughness of hollow slab beams using a prestressed steel hinge line and polyurethane cement composite, and the failure mechanism of test beams under load. By reducing the size of the model beam, the cost was reduced during the test. The stated goal was to provide a basis for the subsequent reinforcement of a hollow slab solid bridge.

## 2. Test Overview

### 2.1. Test Materials

The design strength grade of concrete was C50, and the average compressive strength of cube specimens was 52.6 MP. Four longitudinal reinforcement bars were arranged in the compression zone and tension zone of the hollow slab beam, and a stirrup was arranged every 15 cm along the longitudinal reinforcement bars. Two support reinforcement bars were arranged at each end of support. The polyurethane cement composite material used for toughening the test beam was composed of isocyanate, modified polyether, cement, defoaming agent, catalyst and other components, as shown in Table 1. The density of the test block formed by this ratio was 1400 $kg/m^3$.

**Table 1.** Composition and mass proportions of polyurethane cement composite.

| Composition | Proportion (%) |
| --- | --- |
| Isocyanate | 30.5 |
| Modified polyether | 35 |
| Cement | 33.2 |
| Defoaming agent | 0.5 |
| Catalyst | 0.8 |

Polyurethane toughening test with carbon fiber and glass fiber: see Figure 1. Performance indexes are shown in Table 2.

The type, size, mechanical index and other parameters of the steel strand used for toughening the hollow plate beam, are shown in Table 3.

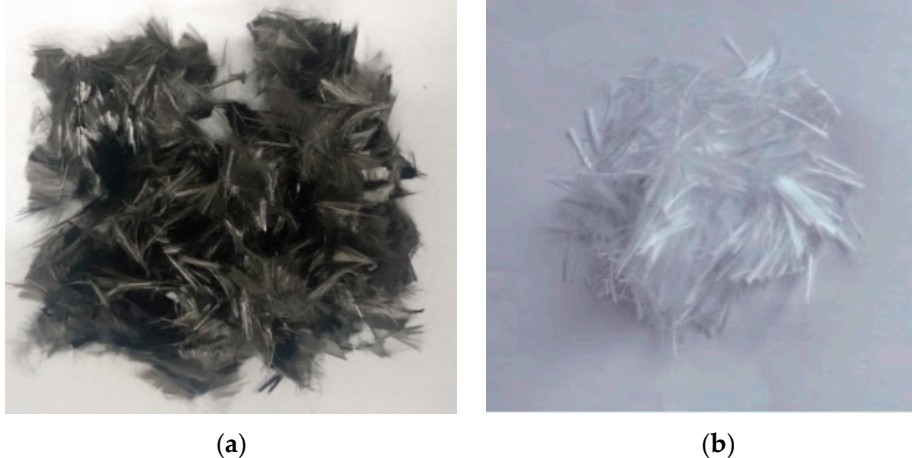

<div align="center">(<b>a</b>)            (<b>b</b>)</div>

**Figure 1.** Test fibers: (**a**) carbon fiber, (**b**) glass fiber.

**Table 2.** Fiber material properties.

| Fiber | Fiber Performance Index | | | |
| --- | --- | --- | --- | --- |
| | Diameter (μm) | Length (mm) | Density (g/cm$^3$) | Elastic Modulus (GPa) |
| Carbon fiber | 4 | 12 | 1.8 | 280 |
| Glass fiber | 9 | 12 | 2.5 | 73 |

**Table 3.** Specifications and parameters of steel hinge lines.

| Category | Nominal Diameter (mm) | Effective Cross Section (mm$^2$) | Design Value of Tensile Strength (MPa) | Elastic Modulus (GPa) | Poisson's Ratio |
| --- | --- | --- | --- | --- | --- |
| 1 × 7 standard | 15.2 | 140 | 1860 | 0.195 | 0.3 |

### 2.2. Design of Test Beam

A total of 4 test beams, numbered L0–L3, were formed in this test. The control beam L0 was not strengthened, and the test beams L1–L3 were toughened in different ways. The length, width and height of the test beams were 300, 50 and 40 cm, respectively. Circular channels with a diameter of 25 cm were arranged in the cross section of the hollow slab beam. Hollow slab beams at both ends of the pouring support measured a support size length, width, and height of 50, 15, and 10 cm, respectively; in the test beams L0–L3, reserved tension channels were supported for the insertion of steel strands. The specific test beam sizes are shown in Figures 2 and 3.

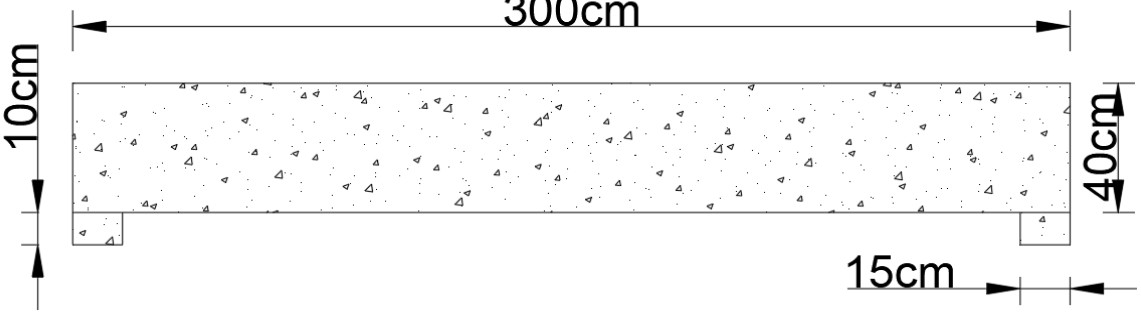

**Figure 2.** Longitudinal section of the test beam.

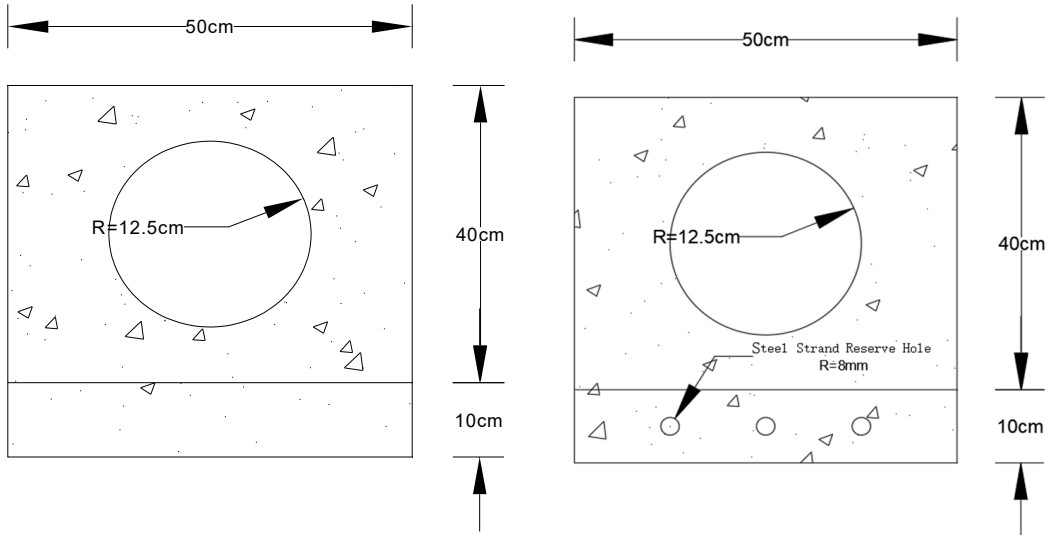

**Figure 3.** Cross section of the test beam.

### 2.3. Toughening Scheme

In order to study the failure phenomenon, bearing capacity and failure mechanism of hollow slab beams under load, the pouring temperature and thickness of polyurethane cement composite should be strictly controlled in the reinforcement process. The same is true of the type of steel strand, reinforcement ratio and tensile stress, among other factors. In this experiment, the toughening material was poured in the room at an ambient temperature of 20 °C.

See Table 4 for the reinforcement scheme of test beams L0–L3.

**Table 4.** Reinforcement scheme of test beams L0–L3.

| Test Beam No. | Quantity of Steel Strand (Root) | Tension (MPa) | Reinforcement Material | Fiber Add | Thickness of the Material (cm) |
|---|---|---|---|---|---|
| L0 | 0 | 0 | / | / | / |
| L1 | 3 | 300 | Steel strand, polyurethane cement composite material | / | 4 |
| L2 | 3 | 400 | Steel strand, polyurethane cement composite material | / | 4 |
| L3 | 3 | 300 | Steel strand, polyurethane cement composite material | 0.04 and 0.04% carbon fiber and glass fiber | 4 |

After the material was cured for 14 days, flexural bearing capacity testing of each hollow slab beam was carried out. Test beam L1 was compared with beam L0 (without toughening treatment), and test beams L2 and L3 were, respectively, compared with test beam L1 in order to observe the failure phenomenon of each test beam under load and evaluate their bearing capacities.

### 2.4. Measuring Point Arrangement and Loading Scheme

In order to better reflect the strain situation along the beam height in the middle of the test beam span under all levels of load, a strain measuring point was set up on the middle side of the test beam span, with an interval of 6.6 cm between two adjacent measuring points. A total of 6 measuring points were set up on test beam L0, and 7 measuring points

were set up on test beams L1–L3. In order to observe the tensile changes of the beam bottom under various loads, two strain measuring points were arranged at the bottom of the mid-span beam.

In order to more clearly reflect the displacement changes of the test beam at the key section under all levels of load, deflection measurement points were placed at the bottom support, at 1/4 of the test beam and at the middle of the span. A total of 5 deflection measurement points were arranged for each test beam. See Figure 4 for the layout of the strain and deflection measurement points.

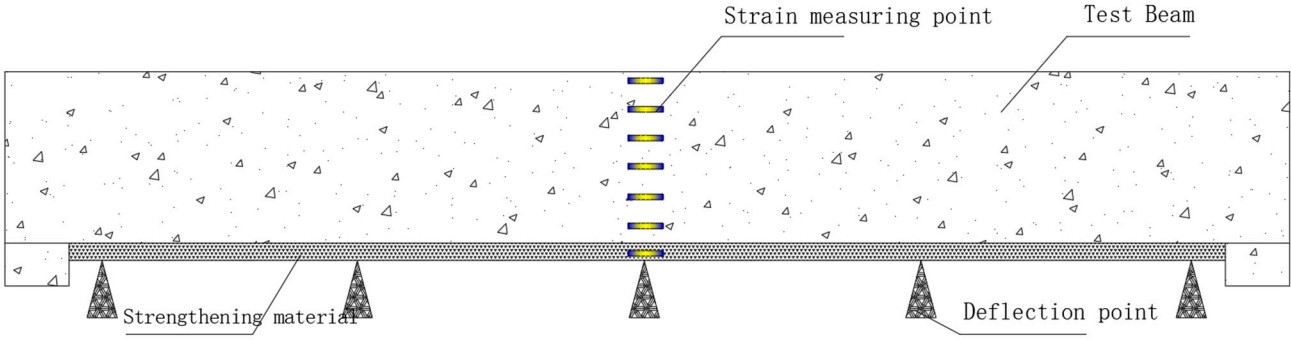

**Figure 4.** Layout diagram of strain and deflection measurement points.

The test used a reaction frame to load through the jack, and the load was applied to the test beam through the steel plate, the pressure sensor, the distribution beam and so on. The distribution beam interval was 25 cm. Before the test, an initial load of 10 KN was applied to observe whether the strain gauge, displacement sensor and pressure sensor were working normally through the static acquisition system. After the inspection, the jack was unloaded, the data of the static acquisition system was cleared, and the formal loading began, with 15 KN loading per level. During the loading process, the phenomena of the test beam under each load were recorded, and a crack width meter was used to observe and record the crack development on the test beams at all times. Additionally, attention was paid to cracking sounds.

## 3. On-Site Reinforcement Processes

In total, 12 steel strands of 4 m length were cut. One end of the reserved channel was anchored with a single anchor head, and the other end was laid with an anchor plate and tensioning jack. The tensioning stress was controlled through the tensioning stress table and elongation. See Figure 5 for tension and anchorage of the steel strands.

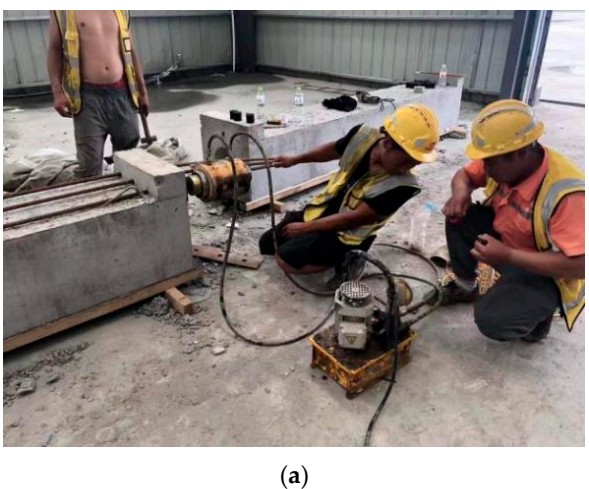
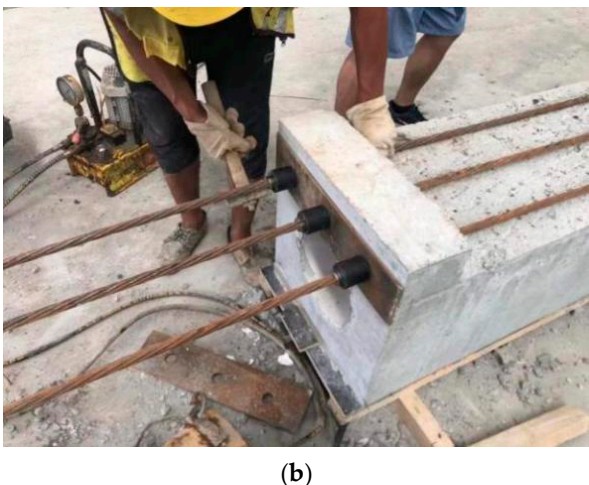

| (a) | (b) |

**Figure 5.** Field photograph of steel strands: (**a**) tensioning, (**b**) anchorage.

Two layers of sponge glue were pasted on the upper edge at 20 cm on both sides of the test beam. In order to prevent the reinforcement material from sticking to the mold during casting and ensure a smooth demold, the side of the template used for casting was wrapped with plastic film, and the treated template was pasted to the side of the beam and firmly fixed with steel nails. After fixing, foam glue was applied evenly to the joint of the template to avoid material spillover during pouring. Before fixing the formwork, the debris on the upper part of the beam was cleaned to ensure that the toughening material was in full contact with the bottom surface of the beam and the steel strand.

The toughening material was weighed and mixed according to the ratio and then poured into the mixing bucket for full stirring with a hand-held agitator. After the materials were mixed evenly, the mixture was poured directly onto the beam surface. Due to the fast condensation of the reinforcement material, it was stirred continuously in the process of mixing and pouring to prevent consolidation of the material. After demolding and curing at 20 °C for 14 days, flexural capacity testing of the hollow slab beam was carried out. The site pouring and loading are shown in Figures 6 and 7.

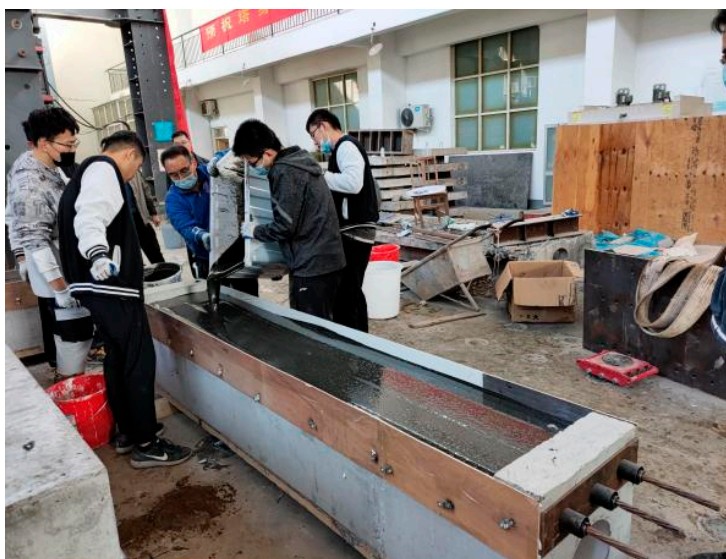

**Figure 6.** Site pouring.

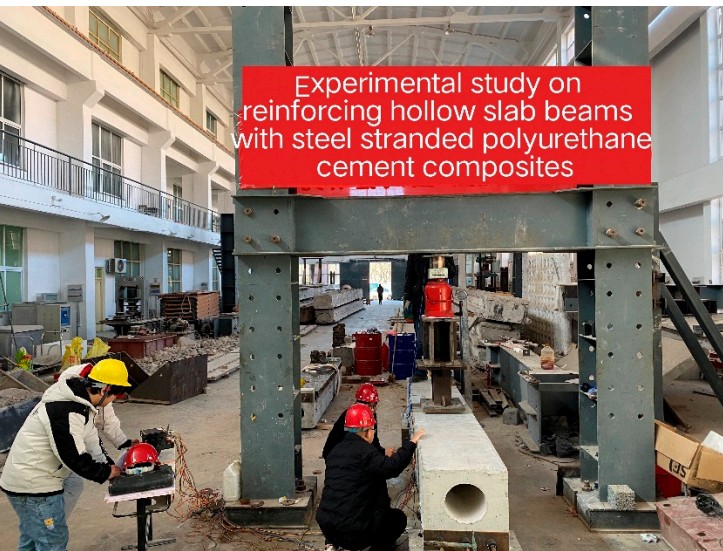

**Figure 7.** Field loading.

## 4. Test Results and Analysis

### 4.1. Experimental Phenomenon

During loading, cracks appeared in all beams, and multiple cracks appeared with the acceleration of deflection growth, rapid crack extension, and crack spread to the top of the beams. Before cracks appeared in L1–L3 reinforced beams, the deflection and strain increased regularly with the increase in load. The deflection of the midspan section increased weakly, and the strain changes at different heights of beams were not obvious. When the crack spread to the top of the beam, it was accompanied by the fracture of the polyurethane-reinforced material. The concentrated forces, corresponding to the midspan sections of L0 and L1–L3 of the reinforced beam at each state, are shown in Table 5. The cracks and fractures of each beam are shown in Figures 8–11.

**Table 5.** Loading state load of each test beam.

| Test Beam No. | Cracks Appear | Multiple Cracks and Increasing Deflection | Rapid Crack Extension | Fracture of Toughened Material |
|---|---|---|---|---|
| L0 | 150 kN | 165 kN | 195 kN | 240 kN (crack propagation beam top) |
| L1 | 225 kN | 330 kN | 370 kN | 450 kN |
| L2 | 315 kN | 330 kN | 450 kN | 495 kN |
| L3 | 240 kN | 300 kN | 380 kN | 465 kN |

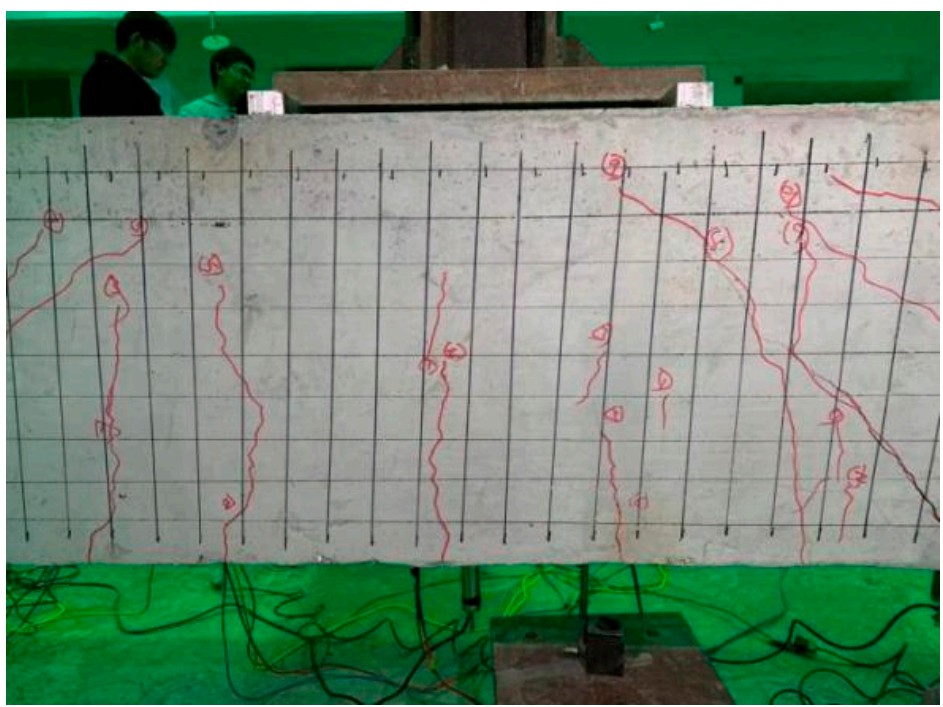

**Figure 8.** Crack pattern of L0 beam.

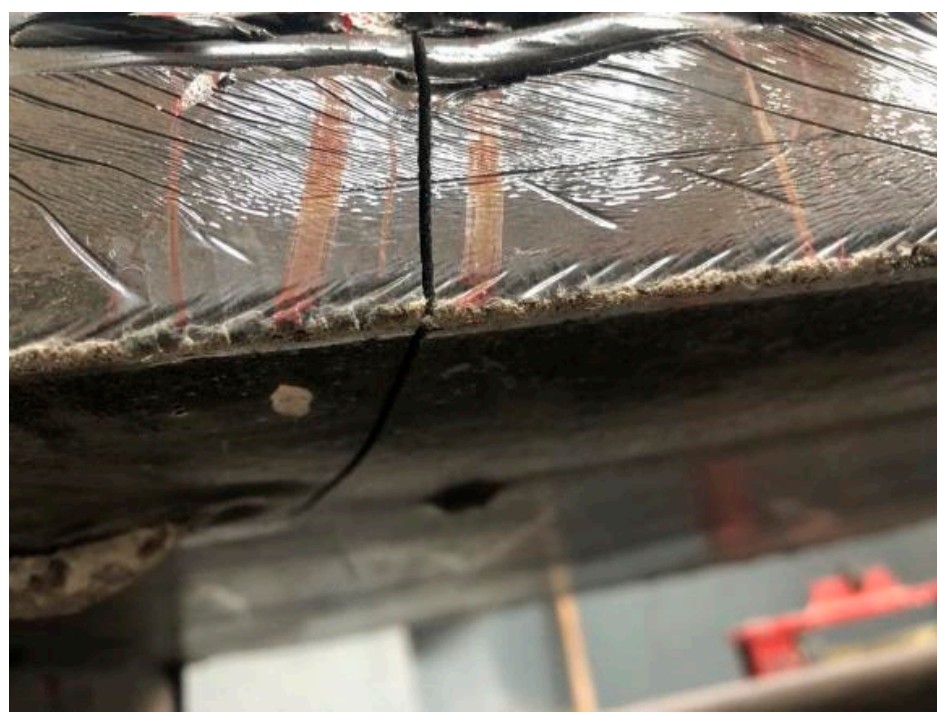

**Figure 9.** Fracture of L1 polyurethane.

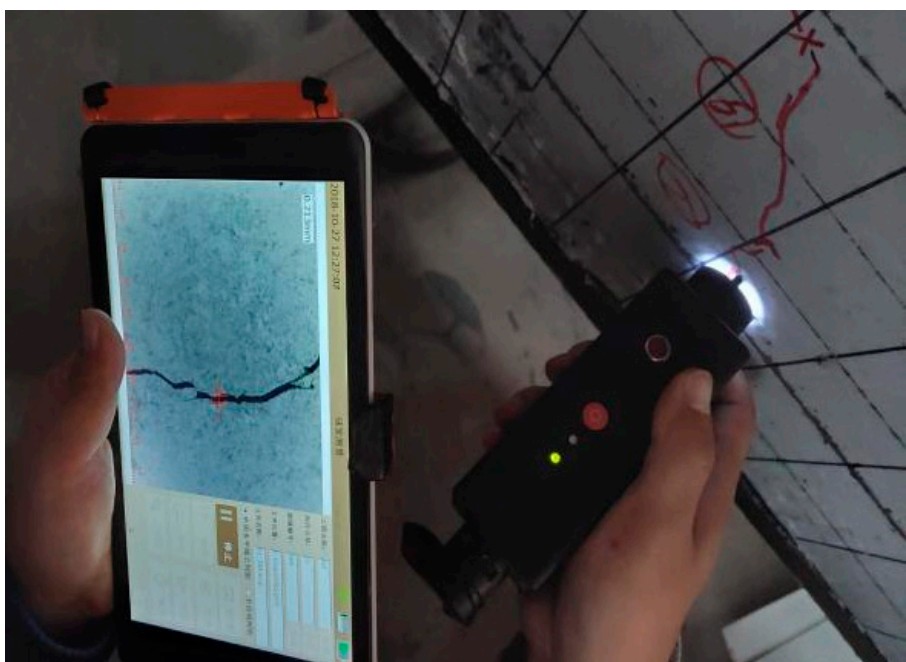

**Figure 10.** Crack opening assessment for L2 beam body.

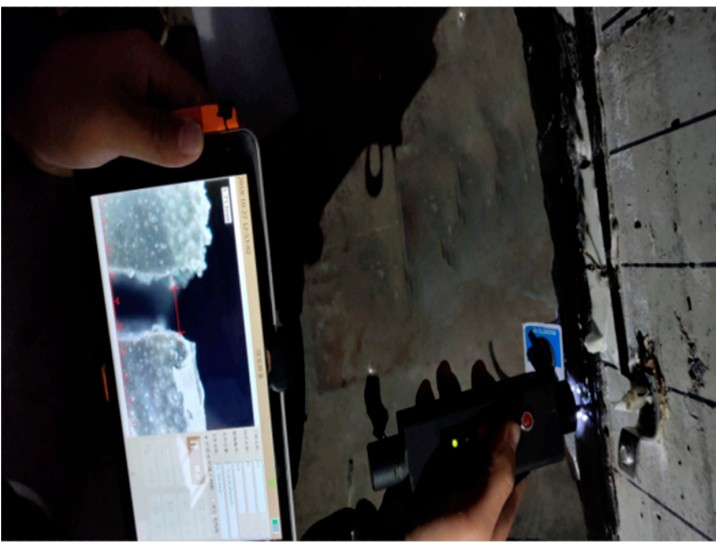

**Figure 11.** L3 crack penetration.

### 4.2. Load–Deflection

The whole process of stressing and deformation of the test beams L0–L3 under load was experienced as follows: no cracks in the concrete, initial crack appearance and development, and finally, failure. When there was no crack in the test beam, the applied concentrated force was lower than the cracking load, the load–deflection curve was basically linear, and the test beam was in the state of elastic force. After cracking, the slope of the load–deflection curve gradually decreased, showing a nonlinear trend. When the bending moment reached a certain value under the load, the deformation of the beam was limited under the constraint of the steel strand, and the crack width and deflection of the test beam increased slowly with the increase in the load. When the test beam was damaged, the deflection increased rapidly, the cracks developed rapidly and spread above the neutral axis, the concrete compressive strain at the edge of the compression zone reached the ultimate compressive strain, the concrete was crushed, the reinforcement layer broke, and the beam side in the middle of the span appeared through cracks. The load–deflection curve is shown in Figure 12.

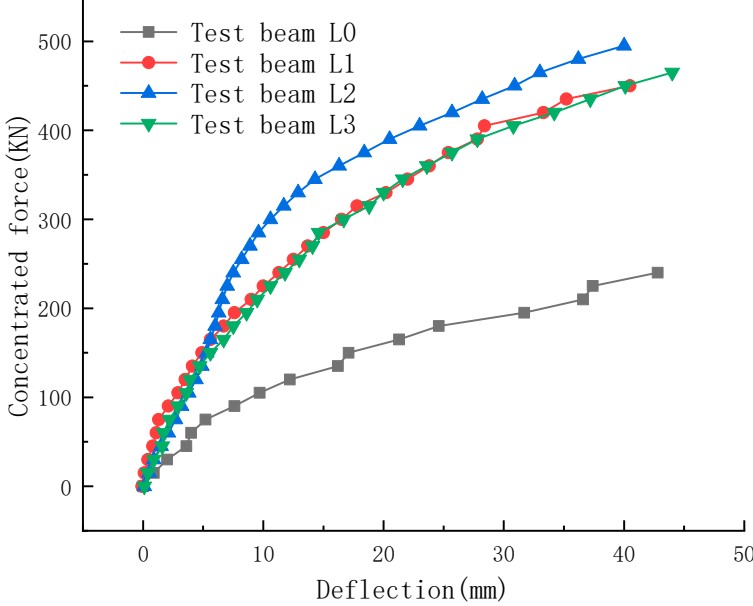

**Figure 12.** Load–deflection curve (crack development yield limit).

### 4.3. Ductility Analysis

The ductility coefficient was divided into displacement, curvature and energy ductility coefficients. In this paper, the prestressed steel strands and polyurethane cement composite material were used to reinforce the bottom of the test beam. The prestressed steel strand had an obvious yield point, so the displacement ductility coefficient was used. The displacement ductility coefficient was used to analyze the resistance ability of concrete test beams L0–L3 to inelastic deformation under load, under the condition that the flexural capacity of test beams L0–L3 did not decrease significantly, which mainly indicated the deformation ability of the test beam at the stage from yield load to ultimate load.

The displacement ductility coefficient μ was calculated by the ratio of the mid-span ultimate deflection $\delta_u$ of the test beam to the yield deflection $\delta_y$. The results of the ductility of the test beams L0–L3 are shown in Table 6, and the scatter distributions of the ductility of each test beam are shown in Figure 13.

**Table 6.** Ductility test results of each test beam.

| Test Beam | $\delta_y$/mm | $\delta_u$/mm | μ |
|---|---|---|---|
| L0 | 31.7 | 42.8 | 1.35 |
| L1 | 25.4 | 40.5 | 1.59 |
| L2 | 23 | 38.6 | 1.67 |
| L3 | 27.8 | 44.7 | 1.60 |

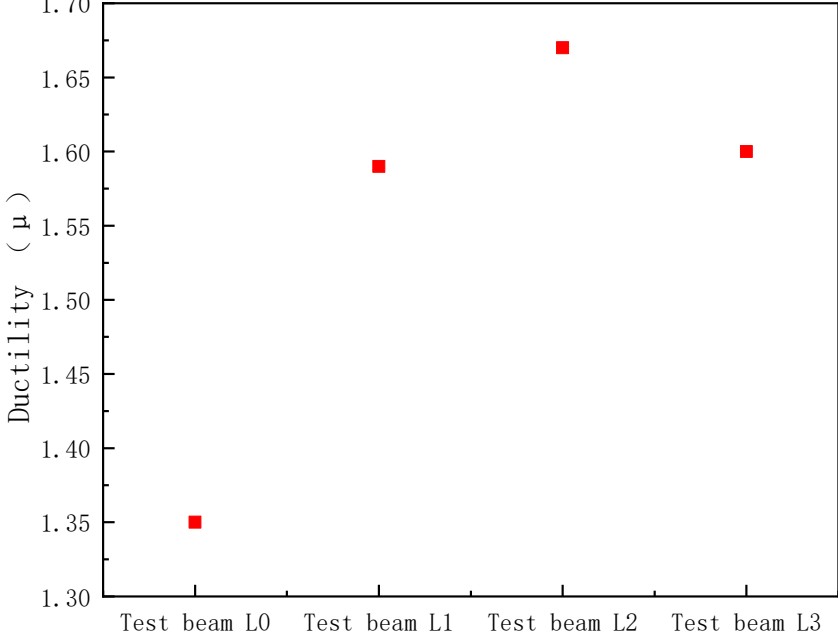

**Figure 13.** Ductility of each test beam.

According to the distribution of ductility scatter in Figure 13, compared with the test beam L0, the ductility levels of the test beams L1–L3 significantly increased, by 17.8%, 23.7% and 18.5%, respectively. This indicated that the combined reinforcement of prestressed steel hinge lines and polyurethane cement composite could effectively improve the ductility of hollow slab beams and provide a greater safety reserve for structural applications. Compared with test beam L1, test beam L2 improved structural ductility by 5% by increasing the tensile stress of each strand by 100 MPa, and the steel strand tensile stress had some effect on the structural ductility. Meanwhile, test beam L3 improved ductility slightly, by mixing 0.04% carbon fiber and 0.04% glass fiber. Therefore, it was speculated that the toughened fiber and polyurethane materials were not fully mixed evenly during mixing, or the density of the toughened fiber was greater than that of the polyurethane cement

toughened material, which sank to the bottom during the mixing process, i.e., part of the fiber stayed at the bottom of the mixing barrel during pouring. By using the combination of prestressed steel strands and polyurethane cement composite reinforcement, the toughness and ductility of hollow slab beams was enhanced, which could provide greater safety to reinforced solid bridges.

### 4.4. Beam Bottom Strain

The variation trend of the strain at the bottom of the test beam was roughly the same as that of the load–deflection curve, and the three stages of stress and deformation of the test beam under load were also verified. As shown in Figure 14, load–beam bottom strain, the test beam was in the elastic stress stage before cracks appeared, and the load–beam bottom strain at this stage was basically linear—that is, with continuous increase in load, the beam bottom strain did not greatly change. At this stage, the ground strand and polyurethane cement composite materials played a small role, mainly in beam force. When the concentrated force reached the cracking load of the test beam, the test beam immediately entered the elastic–plastic stage. At this stage, the load–beam bottom strain curve showed a nonlinear relationship. At this stage, the strain increased significantly with the incremental increase in load. At this stage, steel strand and polyurethane cement composite materials played a major role. When the load reached the yield load, the test beam entered the failure stage. At this stage, the steel bar reached the yield state, and the concrete of the test beam could be crushed or the polyurethane cement composite could be pulled off. The variation trends of L1 and L3 of the test beams were close to that of the load–deflection, which had similar characteristics. The load–beam bottom strain curve could, thus, effectively verify the variation trend of the load–deflection curve and the performance characteristics of the test beam under load, and more scientifically and accurately describe the mechanical properties of the test beam.

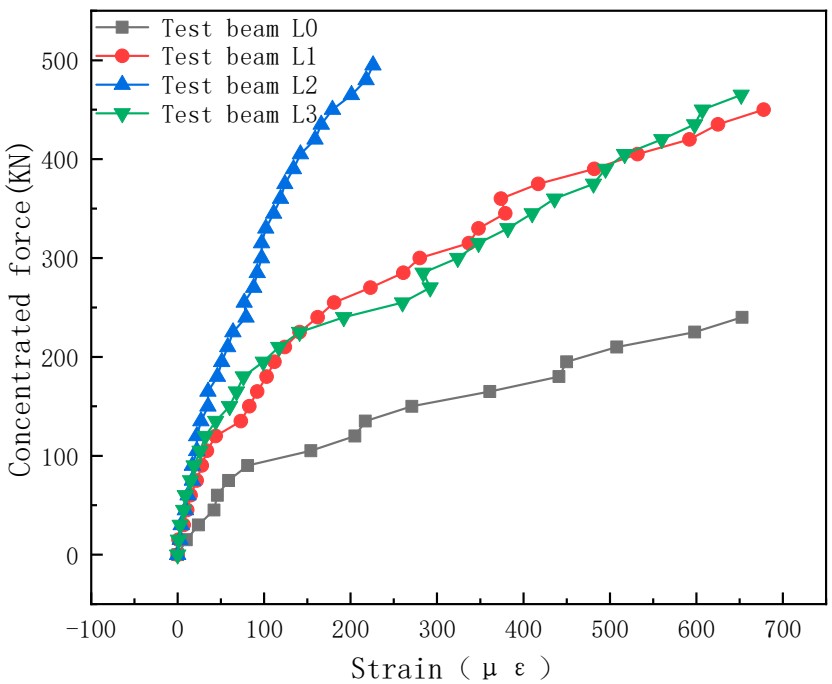

**Figure 14.** Load and strain at the bottom of the test beams.

### 4.5. Strain along Beam Height

The assumption of the plane section was verified by analyzing the strain along the high section of the beam. When the concrete beam was purely bent under the action of the external load, any cross section on the beam rotated around the axis of the beam. At this time, the longitudinal material of the concrete beam did not squeeze—that is, the test beam

did not show concave and convex warping, and the axes of the deformed beam were still perpendicular to each other.

Through the flexural capacity test of the normal section of the test beam, the concentrated force was applied incrementally in the middle of the span. The concrete strain values of different sections of the beam height under different loads were obtained using the static strain collection instrument, and a distribution map of the measured average strain was drawn. As shown in Figures 15–18, when the test beam was in the elastic stress stage, there was no damage to the concrete beam, and the neutral axis of the section gradually moved to the compression zone as the load increased. At this time, the applied load was small, the concrete below the neutral axis was under tensile stress, and the concrete above the neutral axis was under compressive stress. At this time, the strain was an inclined straight line, and the strain distribution along the high section of the beam under the load was a triangle, with the top and bottom opposite. The concrete test beam at this stage fully conformed to the assumption of the plane section. When the test beams in the elastic–plastic stress test (L1–L3) reached the cracking load, below the neutral axis of the concrete cracks, reinforcement with the surrounding concrete occurred, producing relative displacement. However, at this stage, the prestressed steel hinge line and polyurethane cement composite began to undertake the main external load, offsetting part of the tensile stress. Under the action of load, the strain distribution along the high section of the beam was approximately triangular from top to top, and the average strain distribution of other sections of the beam across the crack was still consistent with the assumption of plane section. When the test beam was in the failure stage, the crack width in the tensile zone of the test beams L0–L3 increased sharply and the steel bar yield in the tensile zone occurred in a finite length range. The concrete in the compression zone was crushed within a certain length range, and the polyurethane cement composite did not peel or fracture. The section deformation at this stage was still approximately consistent with the assumption of the plane section.

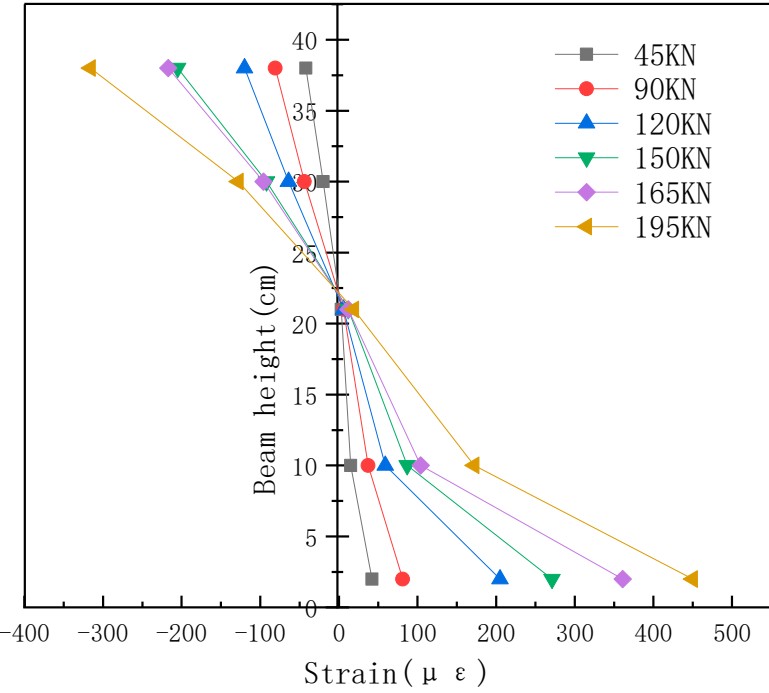

**Figure 15.** Strain of test beam L0 along beam height.

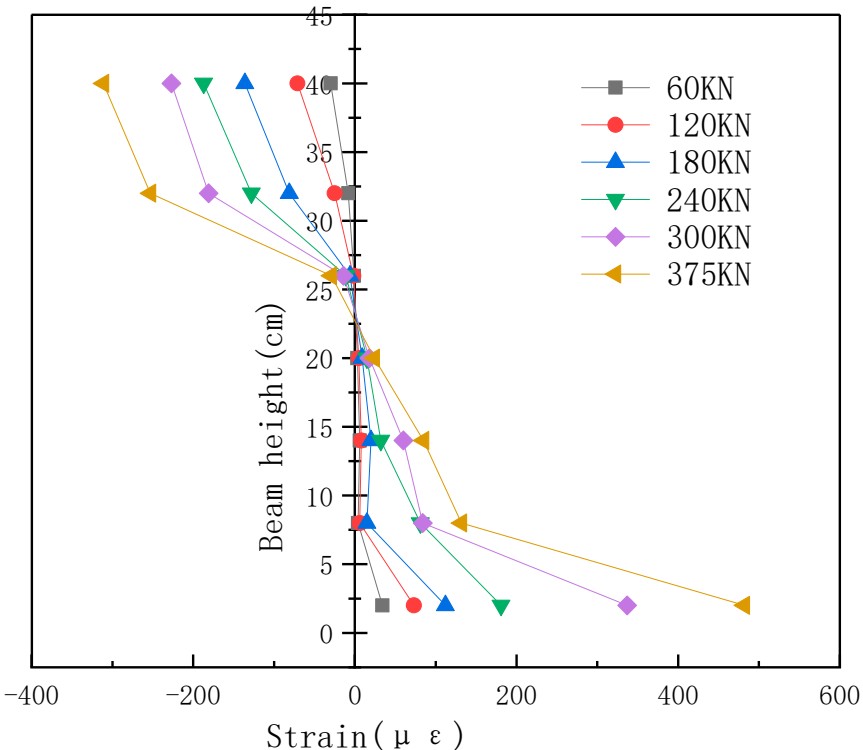

**Figure 16.** Strain of test beam L1 along beam height.

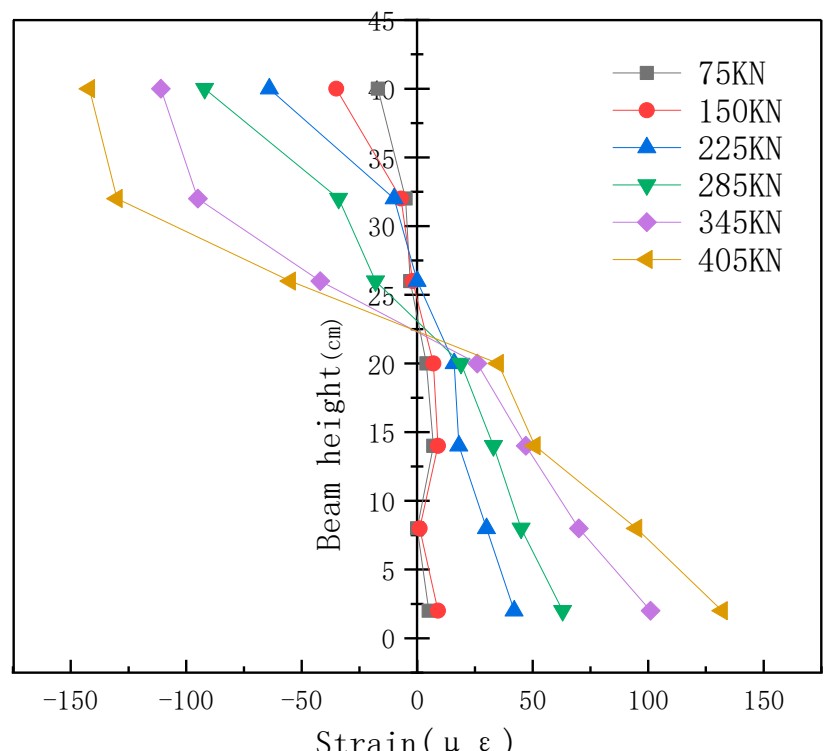

**Figure 17.** Strain of test beam L2 along beam height.

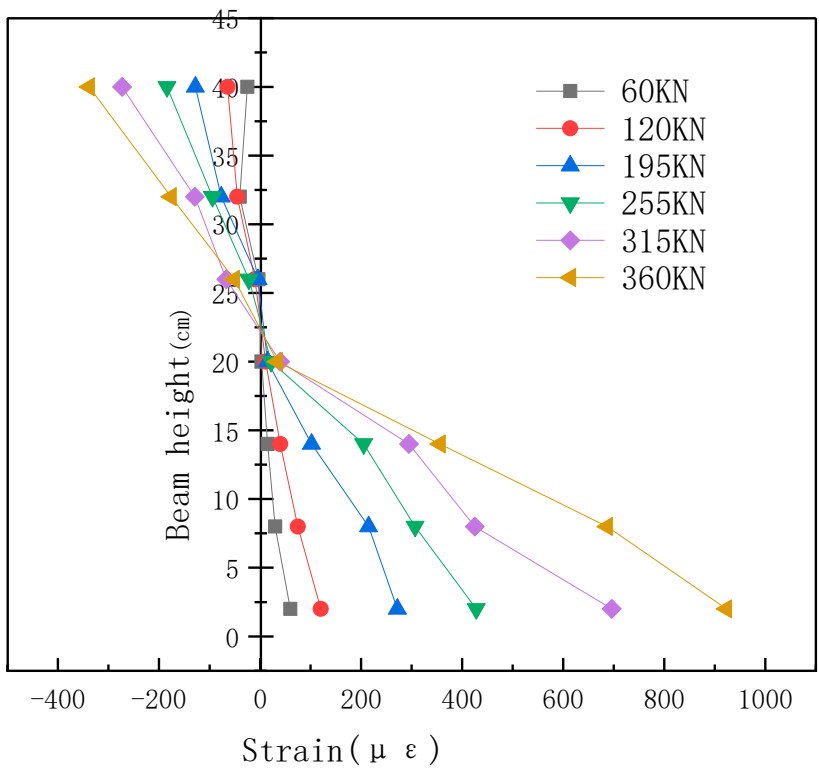

**Figure 18.** Strain of test beam L3 along beam height.

*4.6. SEM Observation*

The good adhesion of polyurethane cement composite could be clearly observed by SEM. There was no gap or crack between the fiber and the polyurethane cement composite, as shown in the figures below. As shown in Figures 19 and 20. It can be seen that the fiber played a positive reinforcement role [14].

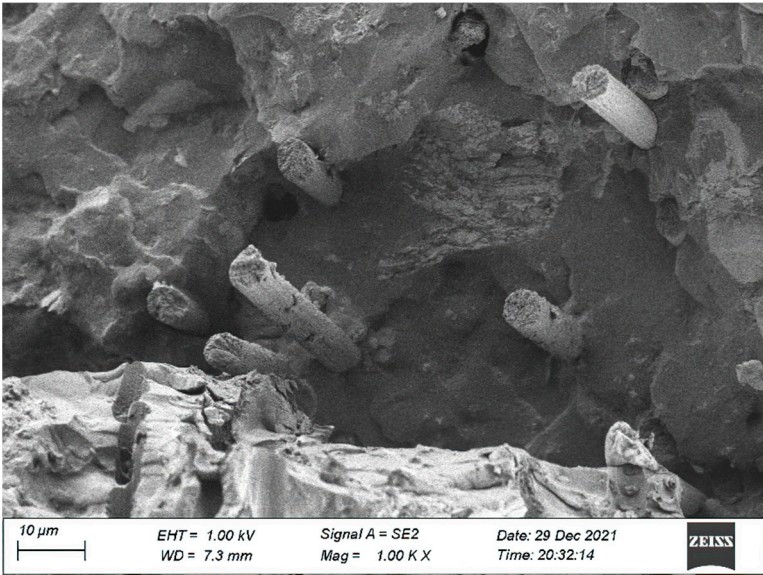

**Figure 19.** A 10 μm SEM micrograph.

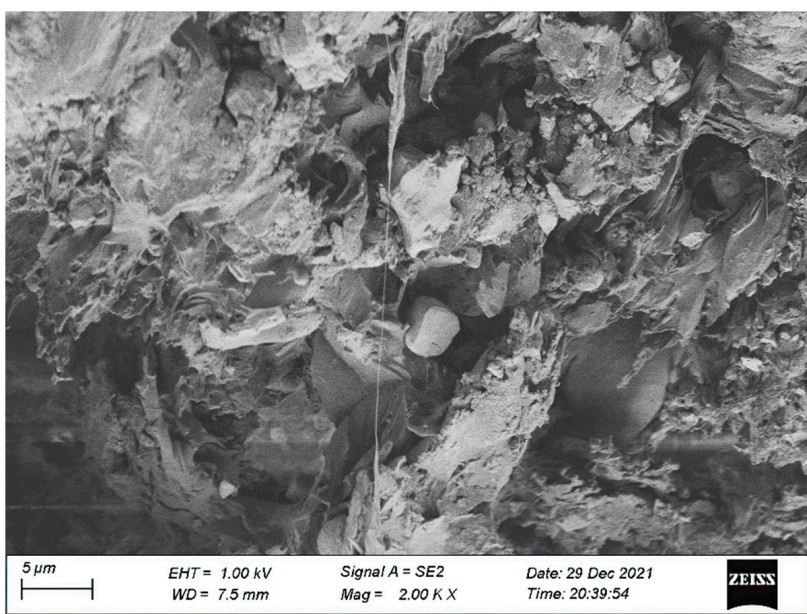

**Figure 20.** A 5 μm SEM micrograph.

## 5. Conclusions

In this study, the failure phenomena, load–deflection curve, ductility, strain at the bottom of the beam and strain along the beam height in the middle of the span were analyzed through the flexural loading of the test beam. The following conclusions were drawn:

(1) Prestressed steel strand polyurethane cement composite material was used for reinforcement, which significantly improved the crack resistance of the test beam. The crack resistance was as follows: test beam L2 > test beam L3 > test beam L1 > test beam L0;

(2) The slope of the load–deflection curve of the test beam without cracks reflected the rigidity of the test beam at this stage. The rigidity of the test beam was as follows: test beam L2 > test beam L3 ≈ test beam L1 > test beam L0. The combined reinforcement of prestressed steel strand and polyurethane cement composite material reduced the deflection of the test beam under load and effectively improved the rigidity of the test beam;

(3) The ductility of test beams L0, L1, L2 and L3 under load was 1.35, 1.59, 1.60 and 1.67, respectively. The capacity of the test beam to resist inelastic deformation was as follows: test beam L2 > test beam L3 > test beam L1 > test beam L0. The combined toughening of the prestressed steel strand and polyurethane cement composite material effectively improved the ground ductility of hollow slab beams, providing greater safety reserves for the subsequent application of bridge reinforcement;

(4) According to the high strain diagram along the beam of each test beam, it was noted that the high strain along the beam of test beams L0, L1, L2 and L3 under load conformed to the assumption of the plane section when the test beam was under elastic stress, elastic–plastic stress and failure stage;

(5) Use of prestressed steel strand and polyurethane cement composite material to strengthen hollow slab beams effectively improved the flexural bearing capacity of the test beam, and this strengthening technology can be further extended to engineering applications.

**Author Contributions:** Conceptualization, Y.C. and J.L.; methodology, D.X.; software, Y.C. and T.Z.; validation, F.C., Y.C. and J.L.; formal analysis, J.L.; investigation, J.L.; resources, J.L.; data curation, Z.L.; writing—original draft preparation, Y.C.; writing—review and editing, J.L. and T.Z.; visualization, X.D.; supervision, H.Z.; project administration, F.C.; funding acquisition, D.X. All authors have read and agreed to the published version of the manuscript.

**Funding:** This study was supported by the Science and Technology Program of Shandong Provincial Department of Transportation (No. 2017B97) and the Key R&D Program of Science and Technology Department of Shandong Province (No. 2019GGX102041).

**Institutional Review Board Statement:** Not applicable.

**Informed Consent Statement:** Not applicable.

**Data Availability Statement:** Not applicable.

**Conflicts of Interest:** The authors declare no conflict of interest.

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
