# Peer review of "Experimental Study on the Bending Resistance of Hollow Slab Beams Strengthened with Prestressed Steel Strand Polyurethane Cement Composite"

_coatings, doi:10.3390/coatings13020458_

Round 1

Reviewer 1 Report

1. I suggest authors to clarify how the dimensions of the test beam is arrived? Is it arrived with the help of scaling method?

2. If applicable, I suggest authors to add discussion points related to cost comparison in introduction part.

Author Response

  1. The size of the test beam is given by the cooperative manufacturer. Under this size, it is arrived with the help of scaling method.
  2. Discussion points related to cost comparison have been added in the introduction.

Reviewer 2 Report

1.       The abstract is very general. A brief discussion on the experimental test results can be included in the abstract.

2.       The key word is given as -steel strand polyurethane cement- kindly check if it is appropriate.

3.       Expand MPC given in introduction- “increase the overall stiffness of bridges by using MPC composite reinforcement”

4.       In this experiment, the toughening material was poured in the room at an ambient temperature of 20 °C- Is the sentence written correctly?

5.       After the material was cured for 14 days- What type of curing was adopted?

6.       The language of the entire  manuscript to be checked-For eg “In order to better reflect the strain situation along the beam height in the middle of 112 the test beam span under all levels of load” , “In order to more clearly reflect the displacement changes of the test beam at the key 119 section under all levels of load”, “The test used a reaction frame to load through the jack, and the load was applied to 126 the test beam through the steel plate”

7.       Explain the rate of loading clearly. It is given as “and the formal 131 loading began, with 15 KN loading per level”- What does the level mean?

8.       “During the loading process, the phenomena 132 of the test beam under each load were recorded”- Explain what phenomena was recorded- Strain or deflection?

9.       In total, s12 steel strands- How many wires were in the strand and what is the diameter of the wires and the strand?

10.   The explanation given is contradicting-“The whole process of stressing and deformation of the test beams L0–L3 under load 182 was experienced, as follows: no cracks in the concrete, initial crack appearance and development and finally, failure”

11.   The sentences have to be organised to explain the behaviour clearly-“ When the test beam was damaged, the 190 deflection increased rapidly, the cracks developed rapidly and spread above the neutral 191 axis, the concrete compressive strain at the edge of the compression zone reached the ultimate compressive strain, the concrete as crushed, the reinforcement layer broke and the 193 beam side in the middle of the span appeared through cracks”- A single sentence has so many explanations.

12.   Spell check to be done- “According to the distribution of ductility scatter in Figure 13, compared with the test 214 beam L0, the ductility levels of the test beams L1–L3 ere significantly increased”

Author Response

1-3.Through inspection, it is appropriate to use steel strand - polyurethane cement composite material. MPC composite is polyurethane cement composite

4. After inspection, the description of the toughening material poured into the mold at the ambient temperature of 20 ° C is correct

5."Material curing for 14 days" adopts 20 ° C on-site normal temperature curing method.

7."15KN load per level" refers to the concentrated force in increments of 15kN per load.

8."Record the phenomenon during the loading process" refers to observing the deflection, crack and other changes of the test beam during the loading process

9. The relevant data of steel strand is given in Table 3

6, 10-12. These descriptions are based on experimental phenomena and data.

Reviewer 3 Report

Coatings

Manuscript ID: coatings_2075788

Experimental Study on Bending Resistance of Hollow Slab Beam Strengthened with Prestressed Steel Strand Polyurethane Cement Composite

by

Li, Cui, Xiong, Dong, Zhang, Cui

REFEREE’S COMMENTS

This is an interesting paper on a subject that should be of great interest to many readers. For location of comments, see the belows.

  1. Title:
  2. It could be more focused/concise.
  3. Abstract:
  4. There is no need to give introductory level information.
  5. The abstract should be supported by quantitative findings.
  6. Introduction:
  7. Use the full name before using its abbreviation.
  8. The authors should refer recently published papers already available in the literature.
  9. Last paragraph: The authors should clearly indicate the originality/novelty of her/his research.
  10. Test Overview:
  11. What is the reason of selecting C50 type concrete?
  12. line 72; what is the reason of selecting15 cm intervals?
  13. Table 1; be consistent with the using capital letters.
  14. Can the authors provide any SEM pictures?
  15. Table 2; "GPa".
  16. Table 3; "1×7 standard"=?
  17. How did the authors selected the dimensions of the beam tested?
  18. line 128; "kN".
  19. Refer relevant papers already published, in order to support the approaches employed here in this section.
  20. On-Site Reinforcement Processes:
  21. The authors are recommended to refer relevant papers published in the literature.
  22. Test Results and Analysis:
  23. line 161-292; The authors are strongly recommended to discuss their own finding with the results of other papers published in the literature. This is particularly very important needs to be completed.
  24. Conclusions:
  25. Use "kN" instead of "            KN" through the submission.
  26. The authors should strengthen the conclusions by referring the quantitative findings.
  27. In General:
  28. Literature review should be extended.
  29. Check out the details of the references cited.
  30. Light-gray-dotted-gridlines would be very useful for the potential readers.

Best regards,

Author Response

The reason for selecting C50 concrete is to refer to the concrete strength of the old bridge being used locally.

The interval of 15 cm is calculated according to relevant specifications

Added SEM micrograph.

KN has been changed to kN

The on-site reinforcement process has been described in "3. On-site reinforcement process"

Round 2

Reviewer 3 Report

-

Author Response

received
